# Temporal-Spatial Evolution and Driving Factors of Global Carbon Emission Efficiency

**DOI:** 10.3390/ijerph192214849

**Published:** 2022-11-11

**Authors:** Ping Cao, Xiaoxiao Li, Yu Cheng, Han Shen

**Affiliations:** 1School of Management Engineering, Shandong Jianzhu University, Jinan 250101, China; 2College of Geography and Environment, Shandong Normal University, Jinan 250358, China; 3School of Foreign Languages, Shandong Jianzhu University, Jinan 250101, China

**Keywords:** carbon emission efficiency, influencing factors, temporal-spatial evolution

## Abstract

With global warming, the continuous increase of carbon emissions has become a hot topic of global concern. This study took 95 countries around the world as the research object, using the Gini coefficient, spatial autocorrelation, spatial econometric model and other methods to explore temporal and spatial evolution, and spatial agglomeration characteristics from 2009 to 2018. The results are as follows: First, global carbon emission efficiency (CEE) showed an overall upward trend, and the average value fluctuated from 0.3051 in 2009 to 0.3528 in 2018, with an average annual growth rate of 1.63%. Spatially, the areas with higher CEE are mainly located in Western Europe, East Asia, and North America, and the areas with lower values are mainly located in the Middle East, Latin America, and Africa. Second, the Gini coefficient increased from 0.7941 to 0.8094, and regional differences showed a gradually expanding trend. The Moran’s *I* value decreased from 0.2389 to 0.1860, showing a positive fluctuation characteristic. Third, judging from the overall sample and the classified sample, the correlations between the influencing factors and CEE were different in different regions. Scientific and technological innovation, foreign direct investment and CEE in all continents are significantly positively correlated while industrial structure is significantly negatively correlated, and urbanization, economic development level, and informatization show obvious heterogeneity. The research is aimed at strengthening exchanges and cooperation between countries, adjusting industrial structure; implementing emission reduction policies according to local conditions; and providing guidance and reference for improving CEE and mitigating climate change.

## 1. Introduction

While economic globalization promotes the rapid growth of the global economy, the global ecological support system is reaching its limit rapidly due to high energy consumption and carbon emissions. Continuously increasing carbon dioxide (CO_2_) emissions are leading to frequent climate anomalies and extreme climate disasters all over the world [1]. Global climate change has become a common challenge facing mankind. There is an approximately linear correlation between the magnitude of global warming and the cumulative emissions of global CO_2_ [2,3]. Since the 21st century, global carbon emissions have grown rapidly, with an increase of 40% from 2000 to 2019. In 2019, global carbon emissions reached 34.36 billion tons, and the per capita carbon emissions reached 4.42 tons per person [4], hitting a record high. Carbon emissions in China, the United States, India, Russia, and Japan accounted for 58.3% of the world’s total. On the whole, global carbon emissions are basically positively correlated with global economic growth [5]. Carbon emission efficiency (CEE) is high or low among the five countries. Except for India, the CO_2_ emissions per unit of energy consumption and the CO_2_ emissions per unit of real GDP in the other four countries have decreased by about 10%, and the CEE is relatively high, indicating that 95 countries in the world are in different stages of economic development, and the differences in energy structure and industrial structure are leading to an internal imbalance of CEE.

Since the 21st century, global carbon emissions have grown rapidly. As BRICS countries (Brazil, Russia, China, India, and South Africa), China, India and Russia accounted for 41.2% of total world carbon emissions in 2014, up from 24.3% in 2000 [6]. Rising global temperatures are a source of increasing discussion and concern. As the global climate continues to deteriorate, glaciers continue to melt, biodiversity is lost, and water cycle models are disrupted, resulting in frequent droughts and severe ecosystem damage [7]. Human economic activities lead to increased greenhouse gas emissions and severe haze pollution [8,9]. In recent years, under the influence of global warming, extreme climate events, such as the melting of glaciers, forest fires, strong hurricanes, extreme cold in winter, and extreme heat waves in summer have occurred frequently [8,10]. The occurrence of extreme events highlights the urgency of taking action to achieve net-zero CO_2_ emissions and adapt to climate change [11].

These catastrophic impacts force all countries and regions to work together to find new solutions to mitigate climate change and reduce carbon emissions [12,13,14]. Therefore, it is necessary to explore the spatial evolution law of global CEE and the heterogeneity of influencing factors. It is of great significance to formulate targeted strategies for energy conservation and emission reduction, tap the potential for emission reduction, and ultimately achieve the goal of global green, low-carbon, and sustainable development.

Since entering the low-carbon economy, the issue of economic growth and carbon emission surge has attracted many scholars. Reducing carbon emissions while promoting economic growth is a key issue that needs to be addressed [15,16,17]. CEE evaluation has become one of the research hot spots in recent years, and previous studies generally defined CEE as the carbon emission per unit of GDP [18]. In addition, some studies have proposed measures such as carbon emission intensity [19], energy intensity [20], carbon index [21] to measure CEE.

At present, the relevant research results of CEE mainly focus on the concept, CEE measurement, and influencing factors. There is insufficient research on the concept of CEE, and a unified view has not yet been formed. Existing studies often explain the connotation of CEE based on a certain industry or field. From an economic point of view, efficiency is defined as the full and most efficient use of limited and scarce resources to meet people’s needs, i.e., to produce more at less cost, given the availability of technology [22]. Carbon emissions refer to greenhouse gas emissions in a broad sense, including CO_2_ and methane. CEE refers to the corresponding effects of social and economic activities that cause carbon emissions [23].

A large number of studies have used different methods to measure CEE in different fields, including industry, manufacturing, transportation, infrastructure and construction. Peng Gao used the super-efficiency relaxation model to measure the carbon emission efficiency of China’s industries under two scenarios: implicit carbon emission and direct carbon emission; Li Jianxuan measured the meta-frontier total factor CEE of 31 manufacturing industries in China from 2012 to 2016, and analyzed the root cause of its low CEE [24]; Based on panel data measurement of various sub-sectors of China’s manufacturing industry, Liu Dongdong analyzed the evolution trend of manufacturing energy carbon emissions and efficiency. On this basis, the convergence of manufacturing energy CEE was tested by using the coefficient of variation and convergence model [25]; Zhao Pengjun used the undesired output-based EBM DEA model and the spatial Durbin model to estimate the CO_2_ emission efficiency of the transportation sector in 30 provinces in China from 2010 to 2016 [26]; Zhang Jingxiao used Hansen’s panel threshold model to analyze the current threshold of environmental regulation on CEE, and studied the impact of transportation costs on CEE of transportation infrastructure [27]; Zhou adopted the three-stage DEA model to evaluate the CEE of China’s regional construction industry [28].

Studies on temporal-spatial evolution have found that the carbon emission in China would continue to grow for a long time [29] and showed significant spatial correlations [30,31]. In addition, some models have been employed to explore the time and space changing features of carbon emission, such as spatial correlation analysis [32], exploratory spatial data analysis [33,34], spatial econometric model [35], and geographically weighted regression model [36].

Compared with existing research studies, the main contributions of this study are as follows: First, based on the research content, the connotation of CEE was defined so as to provide a basis for the construction of a more perfect CO_2_ emission performance measurement system. Secondly, we extended the research scale from urban agglomerations, economic belts, provinces, cities, regions and other regional or relatively smaller scales to the global scale, strengthening the study of global spatial differences in CEE, and providing support for improving CEE in different national conditions. Thirdly, the influencing factors of CEE were systematically studied, and the selection of variables and models were optimized and adjusted according to the expansion of the research areas. In addition, we found that CEE and the efficiency of economic growth have relatively stable relationships, which means that a low-carbon development model can effectively help to sustain economic growth. Current research in different areas has demonstrated the relationship between carbon emissions and economic growth to explain the relationship between the two on an efficiency level, thus providing us with an opportunity to conduct this research.

We organized the rest of this paper as follows: Section 2 reviews the research results related to CEE and discusses our main contributions. Section 3 describes the research methodology and data sources. Section 4 presents the results of our analysis and discusses them. In Section 5, we present our conclusions and countermeasures.

## 2. Research Methods and Data Sources

### 2.1. Research Methods

#### 2.1.1. Measurement of CEE

The paper used a Super-SBM model based on undesired output to measure the CEE of 95 countries around the world. Tone proposed the Super-SBM model based on the SBM model in 2002. It is based on an SBM model to estimate the super efficiency value of decision-making units and is mainly used to solve the problem that the efficiency value of SBM of multiple effective decision-making units is 1, which cannot be ranked. Based on the existing papers, this paper established the global CEE index system from economic development level, labor force and environmental status (Table 1), and is calculated as follows:(1)$$\underset{\;}{\min}\rho^{*}=\frac{\frac{m+1}{m}\sum_{p=1}^{m}N_{p}^{-}/x_{pk}}{\frac{N-1}{N}\sum_{q=1}^{N}N_{q}^{+}/y_{qk}}$$
where *ρ** indicates the relative efficiency value of the decision-making unit; *x* and *y* represent input variables and output variables, respectively; *N_p_*^−^, *N_q_*^+^ represent input and output slack, respectively; *ε_n_* represents the weight vector. The higher the *ρ** value is, the higher the efficiency. In addition, the *ρ** value represents a relative efficiency value, which can only be used for horizontal and vertical comparison of a certain area and cannot fully reflect its true level.

#### 2.1.2. Spatial Autocorrelation

Global space autocorrelation

The global spatial autocorrelation can effectively describe the characteristics of CEE in the whole space and reflects the overall trend of the spatial correlation of CEE in the entire study area. Moran’s *I* index was developed in 1950 by Patrick Alfred Pierce Moran, an Australian statistician. The paper adopted global spatial autocorrelation and reflects the spatial correlation of global CEE through Moran’s *I* index, thereby reflecting the spatial distribution law of global CEE [37]. The model is as follows:(2)$$I=\frac{n\sum_{i=1}^{n}\sum_{j\neq i}^{n}W_{ij}(X_{i}-\bar{X})(X_{j}-\bar{X})}{\sum_{i=1}^{n}\sum_{j\neq i}^{n}W_{ij}\sum_{i=1}^{n}(X_{i}-\bar{X})}$$
where *n* is the number of targeted countries, and *x_i_* and *x_j_* are the CEE values of *i* and *j* countries, respectively. *x* is the average value of CEE values of all countries; *w_ij_* represents the spatial adjacency weight matrix.

2.Local spatial autocorrelation

The paper selects the statistic Getis-Ord* to analyze the local spatial dependence of global CEE, analyzes the correlation of global national units in spatial locations, and reveals the differences between regions [38]. The formula for calculating Ord* is as follows:(3)$$G(d)=\frac{\sum_{j=1}^{n}w_{j}(d)x_{j}}{\sum_{j=1}^{n}x_{j}}$$
where, *d* represents the distance and *x_i_* and *x_j_* represent spatial observations in different regions. *W_ij_(d)* is the weight matrix and *n* is the number of area units.

#### 2.1.3. Panel Regression Model

The IPAT model was first proposed by Enrlich and Holden in 1971 to evaluate the impact of human behavior on the environment. However, this model was too simple, so Dietz and Rosa improved the IPAT model and proposed the STIRPAT model in 1997. This paper explores the influencing factors of global CEE based on the STIRPAT model. The basic expression of the STIRPAT model is [39]:(4)$$I=\alpha P^{x}A^{y}T^{z}E^{c}\beta$$

In Formula (4), P, A, T, and E represent population, economy, technology, and environmental conditions, respectively [40]. *x, y, z,* and *c* represents the estimated parameters of the corresponding factors. *α, β* are constant terms and random error terms, respectively. In order to eliminate possible heteroskedasticity, the formula was converted into logarithmic form:(5)lnI=α+xlnP+ylnA+zlnT+clnE+β

This study considers the influence of economic, technological, environmental, and other factors, and builds a global CEE model:(6)$$lnEI_{i,t}=\mu_{0}+\mu_{1}lnGII_{i,t}+\mu_{2}lnSTRU_{i,t}+\mu_{3}lnFDI_{i,t}+\mu_{4}lnUR_{i,t}+\mu_{5}lnPGDP_{i,t}+\mu_{6}lnIDI_{i,t}+u_{i}+v_{t}+\varepsilon_{i,t}$$

In Formula (6), *EI* is the explained variable-CEE. *GII* represents the level of technological innovation, measured by the global innovation index [41]; *STRU* means the industrial structure, expressed by the proportion of industrial added value in GDP; *FDI* is the ability to attract foreign investment, measured by foreign direct investment; *UR* represents the level of urbanization, measured by the proportion of urban population to the total population; *PGDP* represents the level of economic development, measured by per capita GDP; *IDI* is the level of informatization, measured by the informatization development index; *i* means 95 countries in the world, *t* represents the years 2009–2018; *u_i_* is a country fixed effect, and *v_t_* is a time fixed effect; and *ε_i,t_* is a random disturbance term.

### 2.2. Variable Selection and Data Sources

#### 2.2.1. Variable Selection

Combined with the current status of global sustainable development, the factors affecting CEE were comprehensively considered (Table 2), and the level of scientific and technological innovation, industrial structure, degree of opening-up, urbanization level, economic development level and informatization level were used as explanatory variables to comprehensively explain global carbon emission efficiency [42,43].

#### 2.2.2. Data Sources

This paper took 95 countries in the world as the research area. The relevant data of scientific and technological innovation were mainly derived from the “Global Innovation Index Report” from 2009 to 2018. The relevant data of GNP (USD at present price), carbon dioxide emissions, the proportion of added value of the secondary industry in GDP, the proportion of urban population in total population and per capita GDP (USD at present price) were mainly derived the World Bank database (WB). The per capita oil use equivalent data were mainly derived from the “BP World Energy Statistical Yearbook”. The data related to the information development index were derived from the 2009–2018 “Report on Measuring the Development of the Information Society” released by the European Telecommunication Union. The data on foreign direct investment were mainly derived from the United Nations Conference on Trade and Development Database (UNCTAD). Due to missing data from certain years, the average value or interpolation method of the data of adjacent years was adopted.

## 3. Spatial and Temporal Evolution Characteristics of Global CEE

### 3.1. Time Series Evolution Characteristics of CEE

This paper used the super-efficiency SBM model of Max DEA based on undesired output to obtain the comprehensive technical efficiency (the comprehensive performance of the development level of CEE), pure technical efficiency and scale efficiency of 95 countries in the world from 2009 to 2018 [44,45]. The national average of the three efficiencies from 2009 to 2018, and the time evolution trend during the study period were analyzed (Figure 1). On the whole, the fluctuation range of comprehensive technical efficiency and scale efficiency from 2009 to 2018 was relatively small, showing a clear upward trend, while the fluctuation range of pure technical efficiency was relatively large, showing a downward trend of fluctuation. Specifically, the research period was divided into two stages according to the interspersed trends of comprehensive technical efficiency, pure technical efficiency, and scale efficiency [46]. The first stage was from 2009 to 2014. The fluctuation range and direction of the comprehensive technical efficiency and scale efficiency were basically the same, indicating that during this period, the CEE of 95 countries in the world was more affected by the agglomeration caused by scale expansion [47,48]. The second stage was from 2015 to 2018, and the fluctuation range and direction of the curves of comprehensive technical efficiency and pure technical efficiency were basically the same, indicating that during this period, the CEE of 95 countries in the world was mainly affected by technology driving, and was less affected by scale aggregation.

This paper used the Super-SBM model to measure the efficiency of the panel data of 95 countries from 2009 to 2018. Figure 2 shows the time series evolution trend. Overall, the CEE of 95 countries in the world showed a fluctuating upward trend, rising from 0.3051 in 2009 to 0.3528 in 2018, with an average annual growth rate of 1.62 %. The research period was roughly divided into three stages [21]. From 2009 to 2011 (rapid growth stage), mainly due to the widespread concern and attention of countries around the world on global warming, the strong promotion of energy conservation and emission reduction policies in various countries, the development of green technology, advocacy green production and consumption have slowed down the increase in carbon emissions [49]. From 2012 to 2015 (fluctuation decline stage), on the one hand, some countries were in a period of rapid economic development [50,51], and the total energy consumption and carbon emissions increased sharply [52]. During the study period, the energy consumption of fossil fuels accounted for more than 95% of the total energy consumption in Algeria, Oman, Kazakhstan, Saudi Arabia, Malaysia and other countries; on the other hand, some countries have relatively more fossil fuels such as coal resources, petroleum resources, and natural resources [53,54]. With the large-scale use of fossil energy, the energy structure is unreasonable [55]. The rapid increase in carbon emissions reduced the efficiency of carbon emissions. From 2016 to 2018 (slowly rising stage), the CEE in this stage rose from 0.3284 in 2016 to 0.3528. In order to cope with climate change, the world’s major economies have vigorously developed clean energy and renewable energy, and cleaner production technology, energy development technology and pollution control technology have been gradually improved, increasing the efficiency of carbon emissions [38,56].

Furthermore, to explore the time evolution trend of CEE from the perspective of each continent, the CEE of the six continents generally showed an upward trend, and roughly showed the distribution pattern of Oceania > Europe > North America > full sample > Asia > South America > Africa [57,58]. Europe was higher than the full sample [59], North America and Asia were close to the average level of the full sample during the study period, and the CEE value in Africa was always lower. There was obvious regional heterogeneity in CEE [60,61], which was mainly due to differences in the stage of economic development [62]. Oceania, Europe, and North America had higher CEE, mainly because some developed countries have achieved economic transformation and industrial structure trends. Reasonably, the tertiary industry and high-tech industries accounted for a high proportion. In 2018, the tertiary industry in Luxembourg, the United States, Malta, Cyprus, Switzerland, the United Kingdom, France, and the Netherlands all accounted for more than 70%. With advanced technologies such as carbon technology and resource conservation technology, the level of urbanization and the level of education of residents were relatively high, the concept of sustainable development of green production and life had been basically established, and energy consumption had been reduced, which had a positive effect on the improvement of CEE. The economic development of Asia, South America, and Africa was in the development stage of high pollution, high energy consumption, high emissions, and low incomes [63,64]. The industrial structure and energy structure were unreasonable, and the proportion of secondary industries were relatively high [65]. In 2018, the added value of the secondary industry in Oman, Saudi Arabia and Algeria accounted for 55.18%, 49.54% and 47.89% of GDP, respectively. Most of these rely on traditional industrial sectors with a single structure, which are powered by conventional energy and cause great environmental pollution [66]. Therefore, these countries should give full play to their resource advantages, change the extensive economic development mode, promote low-carbon development, promote the transformation and upgrading of the energy industry, and gradually improve the CEE by strengthening the green upgrading of traditional industries, upgrading the level of green technology innovation, and developing modern energy technologies.

### 3.2. Spatial Evolution Characteristics of CEE

#### 3.2.1. Characteristics of Spatial Differentiation of CEE

According to the calculation results, it was found that the variation trends of the coefficient of variation, the Gini coefficient and the Theil index are basically the same, with the Gini coefficient ranging from 0.7938 to 0.8094 (Table 3), indicating that there is a certain spatial difference in the global CEE, with a polarization trend. In 2018, the difference in CEE among countries was the largest, with the highest difference being between Zimbabwe (1.1927) and the lowest being Tajikistan (0.1135), by 1.0792. The difference was the smallest in 2009, with a difference of 0.8743 between the UK (0.9823) and Mongolia (0.1080). On the one hand, some countries have a good foundation for economic and social development, relying on the advantages of technology, talents, capital, and other advantages to gradually transform to a sustainable development model, and the efficiency of energy utilization continues to improve. On the other hand, differences in geographical locations, resources and environmental bearing conditions, and stages of economic development have existed among countries for a long time, and some countries are affected by these differences in economic development mode transformation, energy utilization efficiency, industrial structure adjustment and optimization, etc. The balanced development of carbon development has caused a certain impact [67].

According to the calculation results of CEE of 95 countries in the world from 2009 to 2018, the study drew the spatial distribution map of countries in 2009 and 2018 (Figure 3) and used the natural breakpoint method to divide them into five categories to analyze the spatial pattern characteristics and laws of efficiency of the carbon emissions of countries around the world.

It has advantages in production technology and selection of energy-saving equipment. It has launched the “Clean Power Plan” and has invested more in renewable resources such as solar energy, wind energy, green hydrogen energy, and nuclear energy. It has accelerated the low-carbon transformation of the energy system. Most of the investment is invested in the tertiary industry with high technology content and low pollution. At the same time, it will reasonably guide the flow of foreign investment, optimize the investment structure, promote the optimization of industrial structure, and reduce energy consumption in key industries, and phase out industries with low output value and high energy consumption, and through policy support and enactment laws and regulations to take relevant measures to improve CEE.

In addition, on the one hand, the Brazilian government reduced agricultural carbon emissions by promoting crop rotation, biological nitrogen fixation, and integrated production of agriculture, forestry and animal husbandry, and other advanced methods. At the same time, it vigorously developed bio-fuels, especially ethanol fuels and related industries, adjusted the structure of energy utilization, and reduced atmospheric emissions. Mongolia, Tazania, Tajikistan, Nicaragua, Mozambique, Nepal, Botswana, and other countries have low CEE, mainly distributed in Africa and the Middle East, because most of these countries are developing countries and have rapid economic and social development. Relying on the consumption of resources and energy, the secondary industry accounts for a large proportion [68]. The industrial structure is dominated by low-efficiency, low-tech, and high-emission industries. The improvement of social productivity is at the expense of environmental damage and resource depletion. One-sided pursuit of economic growth has led to a dramatic increase in carbon emissions. There are significant differences in CEE in different countries. It is necessary to strengthen energy and technical cooperation between countries in various fields, phase out traditional production methods with high energy consumption and high emissions, establish and improve a low-carbon environmental protection system for regional collaborative governance, and vigorously develop circular economy and environmental protection. All countries should vigorously develop a circular economy and low-carbon economy, so as to conform to the international trend of green, energy saving, low-carbon and innovative development, and achieve the goals of green and low-carbon development [69,70].

#### 3.2.2. Spatial Correlation Characteristics of CEE

In order to explore the spatial agglomeration pattern and evolution of CEE of 95 countries in the world from 2009 to 2018, the study calculated the global Moran’s *I* value of CEE of 95 countries in the world. (Table 4).

Global spatial correlation pattern

From the results of the global spatial autocorrelation test, the global Moran’s *I* values in 2009, 2014 and 2018 all passed the significance test, and Moran’s *I* value of the CEE is a positive number. The global CEE showed a significant positive spatial agglomeration effect. From the perspective of the dynamic evolution trend, the Moran’s *I* value showed a slight fluctuation trend of first increasing and then decreasing, and the spatial difference showed a certain stage characteristic [71]. From 2009 to 2014, the Moran ‘s *I* value increased from 0.2389 to 0.2406, and was in the stage of rapid increase, indicating that the global agglomeration of regions with high CEE was increasing, and the agglomeration of regions with low CEE was also increasing, and the spatial development difference had become more and more significant; From 2014 to 2018, Moran ‘s *I* value increased from 0.241 It dropped to 0.1860, in a stage of gradual decline, and the trend of spatial agglomeration weakened [72].

2.Local spatial correlation pattern

Due to the problem of ignoring the potential instability of spatial processes in global spatial autocorrelation [73], and in order to further explore the spatial correlation between a country in the world and its neighboring countries, the research combined Gaoda and ArcGIS software to analyze the study area. The Moran index among countries was calculated and its significance was tested so as to conduct local spatial autocorrelation analysis on the CEE of countries in the study area. It can be seen from Figure 4 that the overall spatial pattern of global CEE was relatively stable, showing a polarized pattern of H-H and L-L agglomeration locally.

From 2009 to 2018, the global carbon emission in H-H zone extended from central Europe to the whole, and together with the spatial scope, continued to expand, and the degree of agglomeration gradually increased, indicating that the CEE in Europe continued to improve. The L-L area is concentrated in southern Africa and Central America. With the change of time, the L-L area evolved from a large-area cluster to a small-scale point, and the spatial extent was shrinking year by year, indicating that the carbon reduction in Africa and Central America had achieved initial results. The overall change in the L-H area was not significant. For example, in Iceland, the spatial distribution is relatively smooth, while the country’s CEE is relatively high, but there is still a certain gap compared with countries with higher CEE, and there is a large room for improvement in the future. The H-L area shows a trend of increasing, the overall fluctuation is relatively large, and its spatial range is increasing.

## 4. Analysis of Influencing Factors of Global CEE

### 4.1. Fitting Analysis of Global CEE and Related Variables

The global innovation index (GII), the proportion of industrial added value in GDP (STRU), foreign direct investment (FDI), urbanization (UR), the per capita GDP (PGDP), information development index (IDI) and global CEE were fitted with a scatterplot (Figure 5), in which there is a relatively obvious linear relationship between technological innovation, foreign direct investment, per capita GDP and CEE. In order to explore the influence of various factors on CEE, further models need to be established to clarify the influence coefficient and direction.

### 4.2. Correlation Analysis and VIF Test

The correlation test results are shown in Table 5. The Pearson correlation coefficient between scientific and technological innovation and carbon emission efficiency is 0.3492, showing a significant positive correlation at the 1% level. The Pearson correlation coefficient between industrial structure and carbon emission efficiency is −0.2172, and the correlation is negative at 1% level. The Pearson correlation coefficient between FDI and carbon emission efficiency is 0.4810, showing a significant positive correlation at 1% level. The Pearson correlation coefficient between urbanization and carbon emission efficiency is 0.3989, showing a significant positive correlation at 1% level. The Pearson correlation coefficient between economic development and carbon emission efficiency is 0.6720, and there is a significant positive correlation at 1% level. The Pearson correlation coefficient between informatization development and carbon emission efficiency is 0.4996 and shows a significant positive correlation at 1% level. VIF test was conducted on each variable in the study. As shown in Table 6, the maximum VIF value was 3.99, the average value was 2.09, and all values are significantly less than 10. Therefore, possible multicollinearity problems could be excluded, and empirical analysis could be conducted.

In order to prevent the “pseudo-regression” in the multiple regression model and ensure the scientificity of the model estimation results, the study used LLC and ADF to test the stationarity of each panel data. (Table 6)

### 4.3. Descriptive Statistics and Stationarity Test

In order to further analyse the variables, descriptive statistics of variables affecting factors of CEE in 95 countries were conducted to make the data more systematic and visualized. As shown in Table 7, the minimum CEE is 0.0965 and the maximum is 1.1928, indicating that there are significant differences in CEE between different countries. There are great differences between the maximum and minimum values of scientific and technological innovation, industrial structure, foreign direct investment, urbanization, economic development and informatization development.

In order to prevent “spurious regression” in the multiple regression model and ensure the validity and scientificity of model estimation results, LLC and ADF were used to test the stationarity of unit roots of each panel sequence. The results found that the data tested by LLC and ADF are more stationary, and have passed the test of significance. It shows that the panel data of the study has strong stationarity and can be further calculated (Table 8) [74].

#### 4.3.1. Overall Result Analysis

The research carried out regression model analysis on the influencing factors of CEE in 95 countries around the world from 2009 to 2018. In order to eliminate the influence of heteroscedasticity on the regression results, the research data underwent logarithmic processing to make the gap between the measured values smaller, and ensure the residual error after calculating the logarithm is the relative error, which is smaller than the absolute error. The random effect model, individual fixed effect model, time fixed effect model, two-way fixed effect model, systematic GMM model, and panel Tobit model were used to perform regression analysis on influencing factors. Combined with the results of Hausman test, a fixed effect model was selected for influencing factor analysis. The used data changed with time, so the individual effect and year effect were fixed, and the regression results of the Fe-tw model were used for analysis (Table 9).

There was a significant positive correlation between scientific and technological innovation and CEE, and the influence coefficient is 0.2492, which has passed the significance test at the 1% level. In 2018, countries such as the United States, Switzerland, and the United Kingdom with higher scientific and technological innovation indices had higher CEE, and the trend of the two showed a high consistency [75]. Through the research and development of low-carbon and energy-saving technologies, including advanced production processes and the recycling of resources and energy, these countries expanded the scope and depth of the development and utilization of renewable resources, promoted the transformation and upgrading of energy-intensive industries, and reduced the consumption of resources and energy in the process of production and life, so as to effectively improve energy utilization efficiency and CEE [76].

The industrial structure had a significant negative correlation with CEE. The influence coefficient is −0.0977 and it passed the significance test at the 1% level. Some countries in the study area were in a period of rapid industrialization. The rapid development of industry has an inhibitory effect on the improvement of CEE. The industry has the development characteristics of low added value, low efficiency, high energy consumption and high emissions at the expense of a large amount of resources. It is characterized by low added value, low efficiency, high energy consumption and high emissions, low efficiency of resources and energy utilization, and a sharp increase in carbon emissions. At the same time, due to the limitations of the economic development stage of various countries and the unreasonable utilization of resources caused by traditional industries, the effect of carbon emission scale expansion was greater than the economic efficiency improvement effect, making the negative impact of industrial structure on CEE more significant.

Foreign direct investment has a significant positive correlation with CEE, and the influence coefficient is 4.3119, which passed the significance test at 1% level. With the continuous advancement of the globalization process, the introduction of high-quality foreign businessmen in various countries can produce a competitive effect and a demonstration effect to a certain extent, stimulate domestic enterprises to carry out endogenous technology research and development and innovation, and actively absorb the advanced production technology, energy saving and emission reduction technology, energy saving and emission reduction technology of foreign enterprises. Management experience and R&D incentive mechanism, etc., improve energy utilization efficiency, and improve the technology to improve CEE.

Urbanization had a significant positive correlation with CEE, and the influence coefficient is 0.0573, which has passed the significance test at the 5% level. In 2018, the world’s urban population reached 55% (World Urbanization Prospects: The 2018 Revision Report), and the urbanization process of most countries had entered a mature stage, and urbanization development strategies were formulated one after another. The transformation and upgrading of facilities and industrial structures to a green and low-carbon tertiary industry, and the enhancement of urban residents’ environmental awareness and the implementation of low-carbon living and consumption patterns contribute to the improvement of CEE.

There was a significant positive correlation between the level of economic development and CEE, and the influence coefficient is 0.5611, which passed the significance test at the 1% level. On the one hand, countries with a high level of economic development can increase capital investment in R&D technologies related to carbon emission reduction, and directly provide necessary resources for the improvement of CEE by investing in factors such as clean energy technology, green and low-carbon logistics, and low-carbon lifestyles, directly providing the necessary financial support for the improvement of CEE. On the other hand, the improvement of the level of economic development will promote the transformation and upgrading of the economy and promote the development of the industrial structure to be advanced, green, and low-carbon, thereby improving the efficiency of carbon emissions.

The level of informatization development had a significant positive correlation with CEE, and the influence coefficient is 0.0886, which passed the significance test at the 10% level. With the rapid development of modern information technologies such as the global Internet+, big data, and artificial intelligence, countries have promoted information-based means in high energy consumption and high energy consumption through information and data sharing in many links, such as energy resource production, consumption, and comprehensive utilization of waste. The application in the polluting industry realizes the automation and intelligence of the production process, improves the utilization efficiency of energy resources and the work efficiency of production, reduces the emission of pollutants, and improves the efficiency of carbon emission.

#### 4.3.2. Analysis of Different Continents

To explore the influencing factors of CEE in different continents, the random effect model (Re) and the two-way fixed effect (Fe-tw) were used to carry out regression analysis on the panel data of each continent. A two-way fixed-effects model and a random-effects model were used in Asia and North America for analysis (Table 10).

There was a significant positive correlation between technological innovation and CEE in different continents, indicating that technological innovation has promoted the improvement of CEE in all continents. This factor had the strongest influence in Europe, which may be because European countries have developed economies and can invest more funds used for energy-saving and emission-reduction process and technological innovation.

During the study period, Switzerland, Finland, Sweden, Denmark, and other countries were in a leading position in R&D investment and grew steadily. The total R&D investment in these countries accounted for more than 3% of the gross national product, relying on scientific and technological innovation. New clean energy can be found, which can improve the utilization efficiency of existing resources, improve the level of harmless treatment of waste, and reduce carbon dioxide and other waste emissions. The industrial structure has an inhibitory effect on the CEE of different continents, but Asia did not pass the significance test because the economic development of some countries mainly relies on energy consumption, and the manufacturing and construction industries are mainly characterized by “high investment, high emission and high pollution”. The industry accounts for a relatively high proportion, and the production process produces a large amount of environmental pollutants.

There was a significant positive correlation between foreign direct investment and CEE in different continents, and all of them passed the significance test at the 1% level. The increase in foreign investment promoted the improvement of CEE, which is the same as the overall regression result. Urbanization was significantly negatively correlated with Asia, South America, and Africa, and significantly positively correlated with North America, Europe, and Oceania, reflecting the complexity and stage of the impact of urbanization on different continents. Most countries in Asia, South America, and Africa are developing countries with a low economic level and are in the stage of accelerating urbanization [77]. With the accumulation of population and economic activities in cities and towns, the continuous improvement of urban infrastructure will increase the consumption of resources and energy. Residents relatively have no low-carbon concept, which has a certain hindering effect on the improvement of CEE. In North America, Europe and Oceania, the urbanization level is relatively high, and the quality effect of cities and towns begins to appear. The environmental awareness of residents is enhanced, and the implementation of low-carbon living and production modes has promoted the improvement of CEE. The level of economic development has a promoting effect on the CEE of different continents, but South America did not pass the significance test. There is a significant positive correlation between informatization and different continents. This factor has a more significant driving effect in Europe and North America, mainly because most countries in Europe and North America focus on the development of quantum computing, high-end chip design and manufacturing, database systems, and the Internet of Things. Technology and other modern information technology industries and emerging environmental protection industries, build an information sharing service platform, and effectively improve labor productivity and resource and energy utilization efficiency [78].

#### 4.3.3. Robustness Test

Considering that there may be missing variables or causal relationships between variables in model construction, the resulting endogeneity problem will lead to the instability of model regression results [74]. In order to ensure the reliability and stability of the regression results, the research used the one-period lag of each explanatory variable as an instrumental variable and used two-stage least squares (2SLS) to test the robustness of different regions (Table 11). The results showed that, except for the fact the effect of technological innovation on CEE in Asia, Africa, and Oceania is not significant, the impact nature and significance level of other variables were basically consistent with the original regression results.

## 5. Conclusions and Countermeasures

### 5.1. Conclusions

The research adopted Gini coefficient, spatial autocorrelation, spatial econometric model and other methods to analyze the spatiotemporal evolution and agglomeration characteristics of global CEE and used a spatial panel data regression model to explore the influencing factors of global carbon emissions to draw the following conclusions.

First, the global CEE generally showed a fluctuating upward trend, rising from 0.3051 in 2009 to 0.3528 in 2018, with an average annual growth rate of 1.62 %. From the perspective of each continent, it roughly presents the differentiation characteristics of “Oceania > Europe > North America > full sample > Asia > South America > Africa”. The spatial differences in CEE were more obvious, and the countries with higher CEE were mainly located in Western Europe, East Asia, North America and other regions. Low-value countries mainly included countries in Latin America and other regions with rapid industrialization and urbanization, as well as regions in the Middle East and Africa with relatively rich mineral resources such as oil and coal.

Second, the variation trends of the coefficient of variation, Gini coefficient and Theil index were consistent. The Gini coefficient increased from 0.7941 to 0.8094. There are certain regional differences in global CEE, and the difference increased the CEE of various countries in the world. There was a significant positive spatial correlation, showing a certain stage characteristic. The CEE of 95 countries has a spatial distribution trend of high- and low-value agglomeration.

Third, from the overall sample regression results, technological innovation, foreign direct investment, urbanization, economic development level, and informatization development levels are significantly positively correlated with global CEE, and industrial structure is significantly negatively correlated. From the regression results of continents, there are obvious differences in the impact of various factors on the CEE of different continents. Technological innovation and foreign direct investment have a positive impact on the improvement of CEE in different continents, and industrial structure has a positive impact on the improvement of CEE in different continents. It has an inhibitory effect, and the impact of urbanization level, economic development level, and informatization on the CEE of different continents has obvious heterogeneity.

### 5.2. Countermeasures

Based on the spatial-temporal evolution characteristics of CEE in 95 countries and the mechanism of scientific and technological innovation, industrial structure and other factors on CEE, in order to further improve global CEE and promote its coordinated development with economic, social and ecological environment, the following countermeasures and suggestions are put forward from the following aspects: 

First, strengthen exchanges and cooperation between countries to achieve green and low-carbon coordinated development. Countries should strengthen exchanges and communication with countries with higher CEE, promote the balanced development of green technologies such as modern energy technology, environmental pollution control technology, recycling technology, energy saving and emission reduction treatment technology, and promote clean energy, green industry high-tech, etc. Effective flow between different countries will encourage each country to combine its own characteristics and accurate positioning, and strengthen the low-carbon emission reduction working mechanism to improve the global CEE and achieve global green and low-carbon coordinated development.

Second, the industrial structure should be actively adjusted to promote green upgrading. Some heavy-industry countries in the study area, such as Russia, Azerbaijan, and other countries, have a trend of decreasing CEE. Enterprises with outdated production capacity should be eliminated, the development of “green” and “low-carbon” industries such as high-tech industries should be promoted, and the green upgrade of traditional industries should be strengthened. By gradually raising the entry threshold for industries with high carbon emissions, countries should accelerate the reform and transformation of traditional high-polluting enterprises in some countries, guiding them to rely on market operations and use economic means to cut carbon and emissions, and continuously promote their industries to adapt to the green and sustainable development of the economy and society.

Third, emission reduction policies should be implemented according to local conditions and guide residents in green and low-carbon consumption. Due to the unique development status of each country, combined with regional differences, the government has formulated a scientific and reasonable allocation plan for carbon emission quotas. Countries in the low-value CEE area mainly focus on changing the economic growth mode and improving energy utilization efficiency; countries in the middle-value area should actively develop low-carbon industries, support the development of strategic emerging industries, and promote industrial upgrading and optimization. Countries in high-value areas should increase their green technology innovation capabilities and technology spillover effects to form a benign green circular economy within and between countries. Governments of various countries need to improve the quality of the population, increase efforts to publicize the idea of green production and low-carbon consumption, and guide residents’ consumption patterns to develop towards sustainable consumption.

## Figures and Tables

**Figure 1 ijerph-19-14849-f001:**
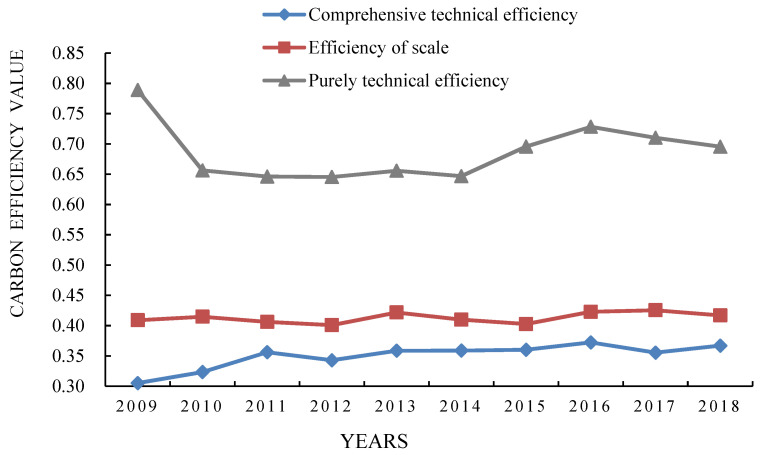
Trend chart of comprehensive technical efficiency, pure technical efficiency, and scale efficiency of global CEE (2009–2018).

**Figure 2 ijerph-19-14849-f002:**
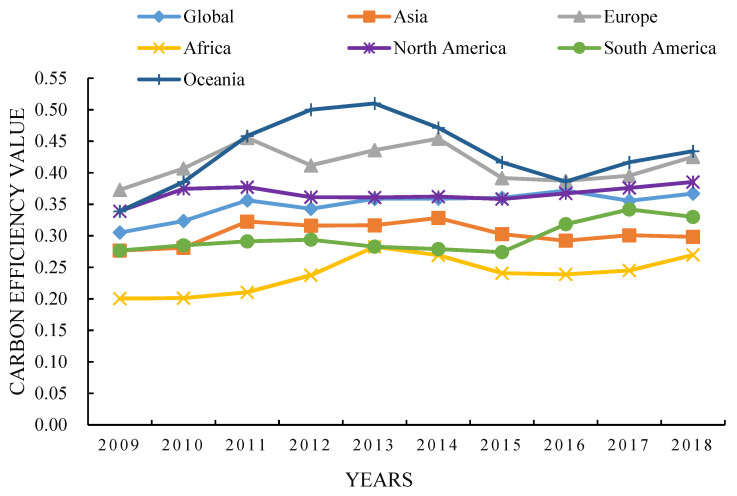
Temporal evolution characteristics of CEE in different continents (2009–2018).

**Figure 3 ijerph-19-14849-f003:**
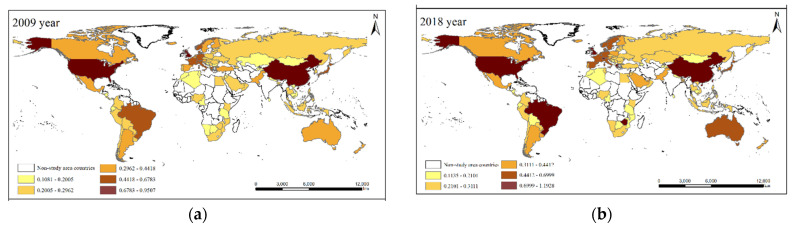
Spatial evolution characteristics of global CEE. (**a**) Description of 2009 year; (**b**) Description of 2018 year.

**Figure 4 ijerph-19-14849-f004:**
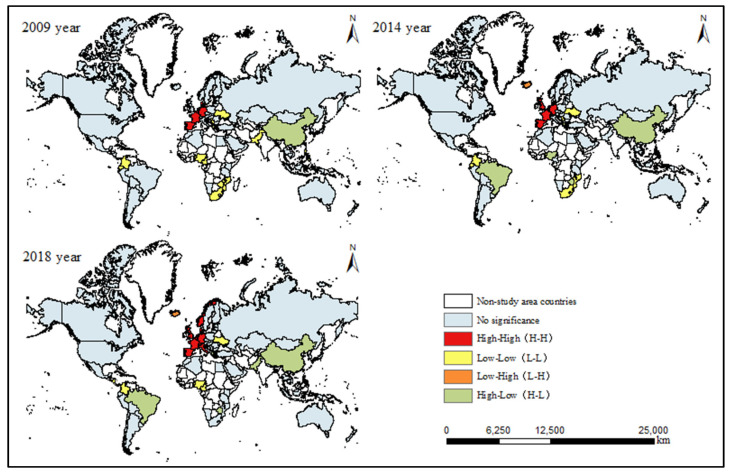
Global CEE Moran’ s *I* spatial distribution.

**Figure 5 ijerph-19-14849-f005:**
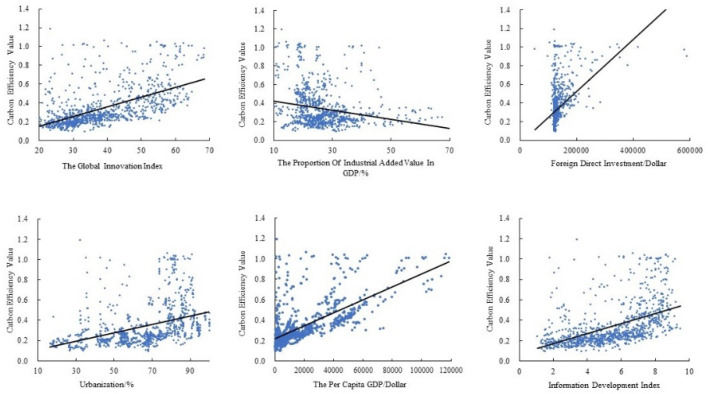
Fitting diagram of CEE and related influencing factors.

**Table 1 ijerph-19-14849-t001:** CEE input-output index system.

Indicators	First-Level Indicator	Secondary Indicators	Unit
Input	Capital	Fixed asset investment	Dollar
	Labor	Labor force	Unit personnel
	Energy	Oil equivalent per capita	Kilogram
Output	Desirable output	GDP	Dollar
	Undesirable output	CO_2_ emissions	Ton

**Table 2 ijerph-19-14849-t002:** Influencing factors of CEE.

Indicators	Index	Explanation	Unit	Model
Explained variable	Carbon efficiency	Carbon Efficiency Value	-	*EI*
Explanatory variables	The level of scientific and technological innovation	Global Technological Innovation Index	-	*GII*
	Industrial structure	Secondary industry as a percentage of GDP	%	*STRU*
	Degree of openness	Foreign direct investment	Dollar	*FDI*
	Urbanization level	The proportion of urban population to total population	%	*UR*
	The level of economic development	GDP per capita	Dollar	*PGDP*
	Information level	Informatization Development Index	-	*IDI*

**Table 3 ijerph-19-14849-t003:** Global CEE regional difference measurement index.

Years	Coefficient of Variation	Gini Coefficient	Theil Index
2009	0.0970	0.7941	0.0587
2018	0.1393	0.8094	0.1641

**Table 4 ijerph-19-14849-t004:** Global autocorrelation of global CEE from 2009 to 2018.

Year	Moran’s *I*	*P* (*I*)	*Z* (*I*)
2009	0.2389	0.0010	4.7322
2014	0.2406	0.0010	4.6180
2018	0.1860	0.0030	3.8930

**Table 5 ijerph-19-14849-t005:** Correlation analysis of variables.

Variable	EI	GII	STRU	FDI	UR	PGDP	IDI
*EI*	1						
*G* *II*	0.3492 ***	1					
*STRU*	−0.2172 ***	−0.1225 ***	1				
*FDI*	0.4810 ***	0.2312 ***	−0.0451	1			
*UR*	0.3989 ***	0.3670 ***	−0.0049	0.2291 ***	1		
*PGDP*	0.6720 ***	0.4753 ***	−0.1890	0.3015 ***	0.6016 ***	1	
*IDI*	0.4996 ***	0.6321 ***	−0.1092	0.2562 ***	0.7131 ***	0.7532 ***	1

Note: ***, denotes significance levels of 1%, 5% and 10%, respectively.

**Table 6 ijerph-19-14849-t006:** VIF test results.

Variable	VIF	1/VIF
*GII*	1.73	0.5783
*STRU*	1.06	0.9427
*FDI*	1.12	0.8957
*UR*	2.17	0.4616
*PGDP*	2.50	0.3997
*IDI*	3.99	0.2504
*Mean VIF*	2.09	

**Table 7 ijerph-19-14849-t007:** Descriptive statistics of variables.

Model	Unit	Min	Max	Mean	SD
*EI*	-	0.0965	1.1928	0.3398	0.2060
*GII*	-	1.97	68.40	32.45	17.85
*STRU*	%	9.98	71.50	26.80	9.01
*FDI*	Dollar	51,687	587,625	133,646.6	35,564.16
*UR*	%	16.43	100	65.18	19.83
*PGDP*	Dollar	428.93	118,823.6	19,114.33	21,860.06
*IDI*	-	1.05	9.475	5.4144	2.0909

**Table 8 ijerph-19-14849-t008:** Stationarity test of panel data.

Model	LLC	*P*	ADF	*P*	Conclusion
*EI*	−10.9549	0.0000	−10.2897	0.0000	smooth
*GII*	−42.4431	0.0000	−6.0822	0.0000	smooth
*STRU*	−8.4160	0.0000	−8.8304	0.0000	smooth
*FDI*	−13.0459	0.0000	−3.6768	0.0001	smooth
*UR*	−11.2808	0.0000	−7.5558	0.0000	smooth
*PGDP*	−8.9553	0.0000	−9.2533	0.0000	smooth
*IDI*	−8.3052	0.0000	−6.3241	0.0000	smooth

**Table 9 ijerph-19-14849-t009:** Regression results of factors affecting global CEE.

Model	Sample Size	Random Effects	Individual Fixed	Time Fixed	Two-Way Fixed	GMM	Tobit
*lnGII*	950	0.0412 (1.30)	0.0619 ** (2.05)	0.1271 *** (3.41)	0.2492 *** (5.17)	0.3171 *** (6.52)	0.0060 (1.22)
*lnSTRU*	950	−0.0778 ** (−2.46)	−0.0674 *** (−2.66)	−0.0842 *** (−5.55)	−0.0979 *** (−6.19)	−0.1220 *** (−3.75)	−0.0775 *** (−4.99)
*lnFDI*	950	−0.5186 (−1.54)	0.1433 (0.44)	4.9787 *** (14.82)	4.3119 *** (12.22)	6.2563 *** (6.90)	4.7257 *** (14.22)
*lnUR*	950	−0.0777 * (−1.91)	−0.1137 (−0.77)	0.2848 *** (11.93)	0.0573 ** (2.44)	−0.0747 ** (−2.34)	0.1063 *** (3.91)
*lnPGDP*	950	0.4277 *** (8.68)	0.5099 *** (8.09)	0. 4785 *** (13.54)	0.5611 *** (8.86)	0.4259 *** (11.67)	0.5099 *** (8.09)
*lnIDI*	950	0.0326 (1.29)	0.1827 *** (4.35)	0.0283 (1.33)	0.0886 * (1.38)	0.0661 *** (−3.00)	−0.0232 (−1.39)
*Cons*	-	2.8778 *** (2.882)	0.2133 (0.26)	−11.4976 *** (−14.24)	−9.8709 *** (−11.43)	−14.7511 (−6.71)	−11.1994 *** (−13.75)
*country fixed effects*	-	-	Yes	no	Yes	-	-
*year fixed effect*	-	-	no	Yes	Yes	-	-
*R^2^*	-	0.5089	0.3881	0.5645	0.6986	-	-
*F*	-	-	30.80	2.12	1.57	-	-
*Log likelihood*	-	-	-	-	-	-	488.3641

Note: ***, **, * denote significance levels of 1%, 5% and 10%, respectively.

**Table 10 ijerph-19-14849-t010:** Regression results of factors influencing CEE in all continents.

Variables	Sample Size	Asia		Europe		Africa		North America		South America		Oceania	
		Re	Fe-tw	Re	Fe-tw	Re	Fe-tw	Re	Fe-tw	Re	Fe-tw	Re	Fe-tw
*lnGII*	950	0.0224 * (1.79)	−0.0418 *** (−3.17)	0.0033 *** (3.81)	0.4610 *** (4.60)	0.0104 (0.90)	0.0191 *** (3.55)	0.0417 *** (3.17)	0.0036 (0.31)	0.0020 (1.54)	0.0520 *** (3.40)	0.0011 (0.35)	0.0417 *** (3.17)
*lnSTRU*	950	−0.0059 (−0.91)	−0.0179 *** (−4.27)	0.0032 (0.34)	−0.3426 *** (−5.21)	−0.2502 ** (−2.10)	−0.2286 ** (−2.37)	−0.2889 *** (−2.45)	−0.0179 *** (−4.27)	−0.0215 ** (−2.36)	−0.0212 *** (−2.99)	0.0723 (1.38)	−0.0180 *** (−4.27)
*lnFDI*	950	0.3400 *** (3.15)	1.0564 *** (5.82)	−0.1414 ** (−2.03)	2.3083 *** (2.66)	9.0996 (0.80)	7.4800 *** (4.74)	7.6983 *** (14.15)	1.0564 *** (5.82)	2.0495 *** (7.02)	2.8994 *** (16.72)	0.3528 (1.09)	1.0564 *** (5.82)
*lnUR*	950	−0.0331 ** (0.76)	0.0207 (0.39)	0.1785 (0.96)	0.3015 ** (2.41)	−0.3835 (−1.61)	−0.2810 ** (−2.13)	1.1434 *** (3.01)	0.0207 (0.39)	0.1574 (0.78)	−0.4767 ** (−2.23)	22.8756 ** (2.23)	0.0207 ** (2.12)
*lnPGDP*	950	0.0669 ** (2.47)	0.5160 ** (2.53)	0.1738 *** (7.41)	2.7660 *** (8.39)	0.6186 ** (2.29)	0.2668 (1.28)	0.7699 ** (2.05)	0.0340 (1.58)	−0.0270 (−0.82)	0.0164 (0.57)	0.0338 (1.58)	0.4257 *** (3.42)
*lnIDI*	950	0.0240 * (2.66)	−0.0978 *** (−3.43)	−0.0422 *** (−3.09)	0.1166 *** (3.22)	0.0117 (0.63)	0.0702 * (2.14)	0.0826 *** (2.80)	−0.0236 (−0.97)	−0.0140 (−0.44)	0.0338 * (1.80)	−0.1334 ** (−2.20)	0.0236 * (2.54)
*Cons*	-	−5.4516 ** (−1.84)	−6.1661 *** (−5.65)	−0.7504 * (−1.88)	−3.9777 *** (−3.97)	−16.6260 (−0.81)	−44.1832 *** (−4.62)	−14.4830 *** (−16.87)	−6.1661 *** (−5.65)	−12.0771 *** (−7.15)	−17.3431 *** (−16.15)	−57.5009 ** (−2.43)	−6.1662 *** (−5.65)
*R^2^*	-	0.3181	0.3073	0.5352	0.7357	0.1335	0.4588	0.9377	0.3073	0.8135	0.8599	0.7774	0.3073
*F*	-	-	1.43	-	1.26	-	3.02	-	1.43	-	2.18	-	1.43

Note: ***, **, and * denote significance levels of 1%, 5%, and 10%, respectively.

**Table 11 ijerph-19-14849-t011:** Robustness test results.

Variable	Overall	Asia	Europe	Africa	North America	South America	Oceania
*lnGII*	0.0022 ** (2.06)	0.0016 (0.67)	0.0045 ** (2.54)	0.0018 (0.85)	0.1852 * (2.04)	0.0034 * (1.96)	0.0011 (0.40)
*lnSTRU*	−0.0198 *** (−7.05)	−0.0148 *** (−4.53)	−0.0331 *** (−3.91)	−0.2728 * (−1.74)	−0.0373 *** (−3.11)	−0.0301 *** (−3.24)	0.0723 * (1.93)
*lnFDI*	1.2021 *** (11.19)	0.9481 *** (4.67)	0.8055 ** (3.76)	7.4871 *** (3.23)	1.2018 *** (10.24)	2.6145 *** (8.57)	1.0882 *** (8.43)
*lnUR*	−0.09996 *** (−4.02)	0.0841 * (1.98)	0.0702 * (2.14)	−0.2520 *** (−3.36)	0.6772 *** (4.61)	0.1334 ** (2.21)	18.3011 *** (3.94)
*lnPGDP*	0.1128 *** (9.67)	0.0591 *** (5.42)	0.1981 *** (8.96)	0.2021 ** (2.41)	0.0657 *** (2.97)	−0.0.627 ** (−2.21)	0.3403 *** (6.98)
*lnIDI*	−0.0325 *** (−3.36)	−0.0492 *** (−3.34)	−0.0644 *** (−2.73)	0.0331 (1.12)	−0.0610 ** (−2.35)	0.0332 ** (1.71)	−0.2108 *** (−8.66)

Note: ***, **, and * denote significance levels of 1%, 5%, and 10%, respectively.

## Data Availability

The data that support the findings of this study are available upon request from the corresponding author.

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
