# Peer review of "Temporal-Spatial Evolution and Driving Factors of Global Carbon Emission Efficiency"

_ijerph, 2022, doi:10.3390/ijerph192214849_

Round 1
Reviewer 1 Report
Review Report on “Temporal-Spatial Evolution and Driving Factors of Global Carbon Emission Efficiency (IJERPH-1989979)” by Ping Cao, Xiaoxiao Li, Yu Cheng and Han Shen
This paper constructs a CO2 emission performance measurement system, which considers urban agglomerations, economic belts, provinces, regions and other small scales. The authors conclude that CEE and economic growth have close relationships. I like this topic. However, I still have some questions:
1. The authors collect the relevant data from the “Global Innovation Index Report” from 2009 to 2018. As we know, most of countries assigned Paris Agreement to control greenhouse gas emissions in 2016. Many developing countries have attached great importance to carbon emissions and technological innovation in recent years. I wonder if your conclusion still holds in practice in recent years or if there will be any new findings.
2. I suggest the author add a correlation analysis table and VIF test to test whether there is a correlation between independent variables and dependent variables in the descriptive statistics section before the regression.
3. I am confusing why the author divides Figure 1 into two stages. It seems to be little difference between the two stages.
4. The titles of Figure 1 and Figure 2 are almost identical, but two figures reflect different content. I suggest the author revise the headings to distinguish each other.
5. I suggest authors to note the number of samples in each model in the table of regression results (Tables 7 and 8).
6. The authors have determined that the two-way fixed effect model is better in Table 7. Why do authors use the random effect model to verify the hypothesis again in Table 8?
Author Response
November 8, 2022
Dear Editor,
Thanks for valuable comments from all experts and referees, which greatly improved the quality of the paper. We have given our revisions and please kindly find the attachment. If there is any improper part of the article, please contact us.
Best regards,
Ping Cao, Ph.D.
Professor, College of Management Engineering
Shandong Jianzhu University
Jinan, Shandong province, 250101, China
Tel: 0086-0531-13553188363
E-mail: 12503@sdjzu.edu.cn

Reviewer 2 Report
This work uses Gini coefficient, spatial autocorrelation and spatial econometric model methods to explore the spatiotemporal evolution and agglomeration characteristics of global CEE by utilizing relevant data of 95 countries in the world, and also uses spatial panel data regression model to analyze the influencing factors of global carbon emissions. The results indicate that the global CEE generally exhibits a fluctuating upward trend from 2009 to 2018, the coefficient of variation, Gini coefficient and Theil index shows a consistent trend, and the correlations between the influencing factors and CEE were different in different regions. In my opinion, this work can be published in IJERPH.
Author Response
November 8, 2022
Dear Editor,
We would like to thank all the experts and referees for their valuable suggestions, which has obviously improved the quality of our paper and also made us learn a lot. If there is any improper part of the article, please contact us.
Best regards,
Ping Cao, Ph.D.
Professor, College of Management Engineering
Shandong Jianzhu University
Jinan, Shandong province, 250101, China
Tel: 0086-0531-13553188363
E-mail: 12503@sdjzu.edu.cn
Reviewer 3 Report
The manuscript entitled "Temporal-Spatial Evolution and Driving Factors of Global Carbon Emission Efficiency", submitted to IJERPH, takes 95 countries around the world as the research object, using the Gini coefficient, spatial autocorrelation, spatial econometric model, and other methods to explore the temporal and spatial evolution and spatial agglomeration characteristics from 2009 to 2018. Some useful countermeasures are given at the end of this manuscript. In general, this manuscript fits the scope of this journal and can arouse the potential interest of the readers. The manuscript might be considered for publication after minor revision. Detailed comments and suggestions are as follow:
1. Line 20: Moran’I -> Moran's I
2. Line 147: give citation(s) here about Moran's I
3. Line 155: give citation(s) here about Ord*
4. Line 161-163: give citation(s) here about IPAT and STIRPAT
5. Table 2: you should mention the table number in the in the main body of the article. Otherwise, it's hard for the reader to locate where this table corresponds to in the manuscript. Please check this problem throughout the manuscript.
6. Section 2.2: If possible, please give the website link or article citation of the data source for the readers to refer to.
7. Section 4.2: Please give more explanation on Descriptive statistics and stationary test? Providing the citation and introduce what it is used for.
8. Line 383: Please give an explanation of LLC and ADF for the reader.
9. Line 390: Please give an explanation on why taking the logarithmic processing can eliminate the influence of heteroscedasticity.
Author Response

(The authors gave the same response as above.)

Round 2
Reviewer 1 Report
Dear editors,
The authors have addressed my all concerns. Thanks.